# Mitigation of Ammonia Emissions from Cattle Manure Slurry by Tannins and Tannin-Based Polymers

**DOI:** 10.3390/biom10040581

**Published:** 2020-04-10

**Authors:** Thomas Sepperer, Gianluca Tondi, Alexander Petutschnigg, Timothy M. Young, Konrad Steiner

**Affiliations:** 1Forest Products Technology and Timber Construction Department, Salzburg University of Applied Sciences, Markt 136a, Kuchl 5431, Austria; thomas.sepperer@fh-salzburg.ac.at (T.S.); alexander.petutschnigg@fh-salzburg.ac.at (A.P.); 2Salzburg Center for Smart Materials, Jakob-Haringerstraße 2a, Salzburg 5020, Austria; 3Land, Environment, Agriculture and Forestry Department, University of Padua, Viale dell’Università 16, 35020 Legnaro (PD), Italy; 4Center for Renewable Carbon, University of Tennessee, Knoxville, TN 37996, USA; tmyoung1@utk.edu; 5Höhere Bundeslehranstalt für Landwirtschaft Ursprung, Ursprungstraße 4, Elixhausen 5161, Austria; konrad.steiner@ursprung.at

**Keywords:** greenhouse gas, NH_3_, tannin-furanic foam, liquid manure, natural polyphenol, agriculture, emission reduction

## Abstract

With the extensive use of nitrogen-based fertilizer in agriculture, ammonia emissions, especially from cattle manure, are a serious environmental threat for soil and air. The European community committed to reduce the ammonia emissions by 30% by the year 2030 compared to 2005. After a moderate initial reduction, the last report showed no further improvements in the last four years, keeping the 30% reduction a very challenging target for the next decade. In this study, the mitigation effect of different types of tannin and tannin-based adsorbent on the ammonia emission from manure was investigated. Firstly, we conducted a template study monitoring the ammonia emissions registered by addition of the tannin-based powders to a 0.1% ammonia solution and then we repeated the experiments with ready-to-spread farm-made manure slurry. The results showed that all tannin-based powders induced sensible reduction of pH and ammonia emitted. Reductions higher than 75% and 95% were registered for ammonia solution and cattle slurry, respectively, when using flavonoid-based powders. These findings are very promising considering that tannins and their derivatives will be extensively available due to the increasing interest on their exploitation for the synthesis of new-generation “green” materials.

## 1. Introduction

Roughly 90% of the ammonia (NH_3_) emissions in Europe are caused by different agricultural systems [1] from which about 41% in the animal sector are emitted by beef cattle [2]. According to Wang et al. [3], the estimated greenhouse gas (GHG) emission of beef cattle is around 50 kg NH_3_ per animal and year. Once in atmosphere, ammonia evolves to nitrogen oxides (NOx), which seriously contribute to the destabilization of the nitrogen cycle with consequent increase of the temperature of the planet [4]. In fact, NH_3_ global warming potential is estimated to be 265 times higher than CO_2_ because it is a precursor of the greenhouse effect and also the ozone layer-depleting gas, nitrous oxide (N_2_O) [5,6]. Other problems associated with ammonia include water pollution or eutrophication [7] and odor nuisance as well as soil contamination and acidification [8].

By 2030 the ammonia emission from agriculture within the EU25 should be reduced by 30%, as decided by the European Commission in 2005 [9]. Since the reductions registered through 2014, the last years have registered a moderate increase of ammonia concentration again [10]. 

In recent years, many research groups focused on the mitigation of ammonia emission in manure. The most common ways include:Manipulating the animal diet with feed additives such as electron receptors, dietary lipids, ionophores, and bioactive plant compounds to reduce the enteric fermentation. It was also considered to lower the crude protein intake, resulting in an overall reduction of NH_3_ evolution [3,11].Covering the manure heap also temporary contributes to the reduction of ammonia emissions, and, hence their oxidation to NOx.Modifying the application method of the manure on the field.Lowering of the pH value of the manure below 5.5, causing a reduction of the ammonia emission by 62% as well as a reduction of methane by 68% [12].Mixing of the manure with different additives, such as urease inhibitor, which blocks the hydrolysis of urea and, therefore, reduces the ammonia emission [13,14]. Other additives, e.g., brown coal, increase the function of the H^+^ ion through cation exchange. Humic acid acts by suppressing the hydrolysis of urea to ammonia [15]. Activated charcoal, pyrochar, or hydrochar have also shown a reduction of ammonia emission due to the adsorption of NH_4_^+^ ions and NH_3_ [4,16,17]. Other extensively studied amendments include inorganic compounds, like lime and coal fly ash [18], alum [19], zeolite [20], or clay [21]. Most of these additives also involve the reduction of the pH.

Tannins are polyphenols which are produced by plants to protect against biotic and abiotic decay [22,23]. They have the strong capacity to complex proteins and this is the most important mechanism exploited to neutralize enzymes of attacking organisms [24,25]. This high complexation capacity is also the reason why tannins are used in vegetal tanning of leather, which is by far the largest application field for these polyphenols [26]. Tannins are gaining attractiveness for other industrial applications because of their sustainability and their high availability: Hydrolysable tannins (Figure 1a) are commonly used in oenology and pharmacology for their astringency [27,28,29,30] and condensed tannins (Figure 1b) are used also as flocculants for water treatment and as substituent for phenol in adhesives [31,32]. This latter family of compound is gaining interest also in polymer science because in recent years several high-performing flavonoid-based products were manufactured. Tannin-based wood preservatives, for instance, have shown an increase in mechanical performance and mitigation of ember time when ignited [33,34]. Tannin-based adhesives have allowed the production of particle boards for interior use with properties comparable with urea-formaldehyde bonded ones [35,36,37] and zero formaldehyde emission [38], while tannin monoliths were applied for carbon-gel production [39,40,41]. In this context, the most attractive products are the tannin foams because, with their strong insulation performance [42], their versatility [43,44], and their outstanding fire resistance [45], they aim to replace synthetic insulation materials such as polystyrene and polyurethane in the green building construction [46,47]. One of the major drawbacks for the tannin foams is that their end-life has not been completely clarified yet. Recently, tannin-foam powders and gels have been used as filter for emerging pollutant with good results [48,49,50,51], so the possibility to exploit the complexation capacity of tannins and their derivatives against ammonia is presented in this article. This feature will open new possibilities for the use of tannin-based polymers after their initial application in building insulation and it also will provide an easy-to-implement alternative to the already existing methods for reducing ammonia emission. Therefore, the aim of this study was to identify the capacity of tannin and tannin-based products to contain the ammonia emissions from liquid solutions, also evaluating the kinetics of the adsorption process. This study might also promote future research in which different tannin-rich agricultural by-products can be considered for the same purpose.

## 2. Materials and Methods 

### 2.1. Chemicals

Chemicals used in the synthesis of the adsorbents were industrial mimosa tannin extract Weibull AQ (Tanac, Brazil), chestnut tannin extract (Saviolife, Italy), furfuryl alcohol (Trans Furans Chemicals, Belgium), sulphuric acid 98% (VWR, Germany), and diethyl ether (Roth, Germany). Liquid manure was collected at Scheiblhub cattle farm in southern Germany.

### 2.2. Adsorbent Preparation and Characterization 

Industrial mimosa and chestnut extracts as representative substances for condensed and hydrolysable tannin, respectively, were used as received. The tannin gel and the tannin-furanic foam were obtained according to previous works [44,52]. In brief, for the foam, mimosa tannin (42%), furfuryl alcohol (26%), water (8%), blowing agent (6%), and sulfuric acid catalyst (18%, diluted to 32%) were mixed and cured in a hot press; for the tannin-gel, mimosa tannin was mixed with diluted sulfuric acid and reacted for 2 h at 90 °C. The tannin-based materials were dried 1 h at 103 °C and then grinded to fine powder (particle size < 125 µm) before being conditioned at 20 °C and 65% relative humidity for 1 day. 

### 2.3. The pH Measurements 

The pH of the 0.1% ammonia solution and that of the manure slurry was measured after adding the adsorbent with an Inolab pH 720 (WTW, Weilheim, Germany) pH meter. The measurement was repeated 3 times.

### 2.4. Adsorption Experiments

Emission of ammonia either from aqueous solution or from liquid manure was monitored in a test chamber, shown in Figure 2. The test setup consisted of a glass box with dimension of 150×150×70 mm³, a petri dish where 1 mL of ammonia-emitting solution was inserted and an ammonia sensor (type MQ-137, Vastmind LLC, Wilmington, NC, USA) sensitive until 500 parts per million (ppm), connected to an Arduino micro controller, which monitored the concentration of free NH_3_ (and other volatile gasses) in the test chamber.

For the preliminary study on aqueous ammonia solution, 1mL of a 0.1% NH_4_OH was placed in the test chamber and the emissions were monitored for one hour. Concentrations of adsorbent of 1%, 5%, and 10% by weight were added to the ammonia solution (e.g., 1 g of ammonia-emitting solution was added of 0.01, 0.05, and 0.1 g of adsorbent, respectively) and the emission was recorded for one hour.

For the adsorption experiments with liquid manure, 7.5% solid content was at first diluted 1:1 with deionized water to allow an effective mixing and homogeneous distribution of the slurry in the petri dish [4]. Also in this case, 1 mL of manure (50%) was added with 1%, 5%, and 10% by weight of adsorbent. The experiments were repeated five times.

### 2.5. Attenuated Total Reflectance Fourier-Transform Infrared (ATR FT-IR) Spectroscopy

ATR-FTIR analysis was done using a Perkin-Elmer Frontier FTIR spectrometer (Perkin-Elmer, Waltham, MA, USA) equipped with an ATR Miracle. Spectra were recorded between 4000 and 600 cm^−1^ with a resolution of 4 cm^−1^ and 32 scans per spectrum. FTIR spectroscopic analysis was conducted in triplicate on the four adsorbents prior and after the experiment. The spectrum of ammonia was taken in a 25% ammonia solution. 

### 2.6. Data Processing and Analysis

Data was analyzed using OriginPro 2019b (OriginLab, Northampton, MA, USA). Fitting and modelling of curves was conducted using the nonlinear fit function of OriginPro 2019b applying the logistic fit.

## 3. Results and Discussion

### 3.1. Adsorption from Aqueous Ammonia Solution

The preliminary study on templated ammonia solution showed that all tannin-based adsorbents significantly contributed in the reduction of the NH_3_ emissions (Table 1).

The pH-value and the emissions were reduced proportionally to the amounts of adsorbent applied.

Despite the addition of chestnut extract decreasing the pH more consistently, the higher ammonia adsorption was observed with tannin gel. In Figure 3 the relation between pH and ammonia adsorption is reported.

We observed that there was a linear proportionality between pH and relative adsorption but the chestnut tannin extract depicted a gentler slope, meaning that its ammonia fixation was not as effective as the other three adsorbents. In particular, the tannin foam powder seemed to be the more effective at lower pH, even if the tannin gel reached higher absolute adsorption levels because of its higher acidity. The mimosa powder showed a similar trend suggesting that the type of flavonoid was the most important factor for the fixation of ammonia.

The FTIR spectrum of mimosa tannin can be interpreted according to [53]. In this study, the powder was measured before and after adsorption (Figure 4) and highlighted that the adsorption of ammonia significantly modified the profile of the spectrum presenting three new absorbances: (1) In the region in red between 1700 and 1630 cm^−1^ the H-N-H scissoring vibration of ammonia occurred, while (2) in the area between 1200 and 800 the N-H wagging took place [54], and (3) the clear band signal at around 1350 cm^−1^ might be due to the complexation of amino groups with tannin [55], with similar mechanism occurring for vegetable leather tanning [56]. 

These evidences would suggest that fixation of ammonia occurs with a process similar to the transformation of hide to leather during vegetable tanning. Although this phenomenon does not necessarily involve new covalent bonding, the complex tannin-ammonia is highly stable. Such complexation guarantees that nitrogen remains long enough in the soil, where it contributes to the fertilization also after the oxidation of ammonia to nitrites and nitrates [57].

### 3.2. Results of the Adsorption Experiment for Liquid Manure

The fixation of ammonia by tannin-based powders occurred successfully, but the applicability of the system had to be tested in real case. Therefore, the adsorbents were applied on 1:1 diluted manure slurry suspension, and the adsorption are registered in Table 2.

In the case of manure, the emitted ammonia was 30 times higher than in the preliminary test with the template but the efficacy of the tannin-based adsorbents was also outstanding. 

In these conditions, the buffering effect of the manure did not allow consistent pH reductions but the extreme efficacy of tannin as adsorbent was confirmed. These findings also corroborated that flavonoids offer higher ammonia adsorptions than the hydrolyzed tannin. Mimosa-based adsorbent showed superior adsorption, always over 97.5%. 

Interestingly, the relative ammonia adsorptions by percentage registered in the cattle manure slurry were significantly higher than the ones obtained by the adsorption of ammonia alone (Table 1). This observation suggests that the efficacy of the adsorption was improved when higher concentrations of ammonia were applied and that the presence of other substances in the manure did not affect the mitigation effect of tannins. On the other hand, we can also highlight that when the ammonia concentrations were low, tannin-based adsorbents could not completely fix them (at least, using the tested concentration of max 10%) and few ppm of ammonia were emitted anyways.

The sensor was built to be selective for ammonia. However, ethanol, CO, and CH_4_ may interfere with the measurement to some extent. In case this interference occurs, the ammonia value registered by the manure reference could have been too high, because part of the emission was due to the other gases. However, if this interference affected the measurements proportionally, the adsorption results in percentage would be the same. If not, the outstanding adsorption performance of tannin-based powder will be not specific, but still highly effective.

### 3.3. Emission Kinetics of Ammonia from Liquid Manure

Monitoring the emission during one hour, we observed that initially the slope of emission was very steep and, after around 30 min, the slope become milder until reaching a plateau after one hour (Figure 5). 

As soon as any adsorbent was added, the emissions of ammonia sank and the plateau of emission was reached sooner (around 20 min for 1% and roughly 15 minutes more for 10% concentration). These fast adsorptions allowed us to add the adsorbent just before field spreading of the slurry because relatively short storage time was needed. The tannin-based adsorbent could be added directly in the tractor tank when the slurry was loaded from the digestor. This process will guarantee a homogeneous distribution of the adsorbent and the higher viscosity will maintain the insoluble tannin fraction in suspension without need for further stirring during the spreading step. 

Modelling of the logistic function (Equation (1)) to fit the experimental data for mimosa tannin and gel resulted in the parameters presented in Table 3, where A_1_ and A_2_ are the calculated starting and finishing values to best fit the experimental data, x_0_ describes the inflection point, and p the power of the function. For liquid manure, the simulated function matched the experimental data with an accuracy of more than 99.99%. For all other simulations, correlations above 90% were observed.
(1)$$f\left(x\right)=A_{2}+\frac{\left(A_{1}-A_{2}\right)}{1+{\left(\frac{x}{x_{0}}\right)}^{p}}$$

Using the modelled equation, the rate of gaseous emission can be calculated. It was found to be quite slow for the manure mixed with adsorbents (between 0.002 and 0.05 ppm/sec at 600 seconds for 10% and 1% gel, respectively). While for the untreated manure the rate of emission ranged from 0.12 ppm/sec (at 200 s) to 0.27 ppm/sec at 600 s. 

Assuming that there was no more significant ammonia emission when the experimental maximum value (allowing a 95% confidence band) was reached, one can use this equation to calculate the exact time. In the case of the manure, the calculated value was at around 1 h. When looking at the 10% gel or mimosa tannin addition, the theoretical value was around 20 minutes and for the 1% adsorbents, after 45 minutes. These values were in line with the experimentally observed ones.

## 4. Conclusions

In this research the mitigation effect of tannin-based adsorbents onto ammonia and gaseous emission was investigated. Chestnut and mimosa industrial tannin extracts, tannin gel, and tannin-furanic foams were tested against a 0.1% template ammonia solution and 50% diluted cattle manure slurry. Significant to outstanding reductions of ammonia emissions were registered in every test. Even small concentrations of tannin-based powders already offer good mitigation of ammonia adsorption. When 10% of adsorbent was used, the emissions of ammonia sank up to 77% for ammonia and 99% for manure. It was confirmed that the pH had an important role on the ammonia adsorption, but the type of tannin also had a major impact. Mimosa tannin-based adsorbents showed roughly 10% higher adsorptions than chestnut powder in both experiments. 

The presence of adsorbent not only decreased the emission of ammonia but also shortened the time until the maximum concentration of ammonia was emitted up to around 10 times (from 30 to 3 minutes).

The FTIR measurements highlighted that ammonia was fixed in the tannin-based adsorbent: The vibrations of free NH_3_ were found in the adsorbent, together with the signal at around 1350 cm^−1^, typical for complexed amino groups in tanned leather, with this latter suggesting a stable complexation of ammonia.

These results promote tannins and tannin-based products as suitable adsorbents to fix ammonia and, hence, to fulfil the requirements of the European Union concerning the reduction of agricultural ammonia emissions. Further studies are necessary to investigate the long-term effects on the ammonia emission reduction and to clarify the transformation process of nitrogen occurring in the soil when complexed to tannin-based adsorbents.

## Figures and Tables

**Figure 1 biomolecules-10-00581-f001:**
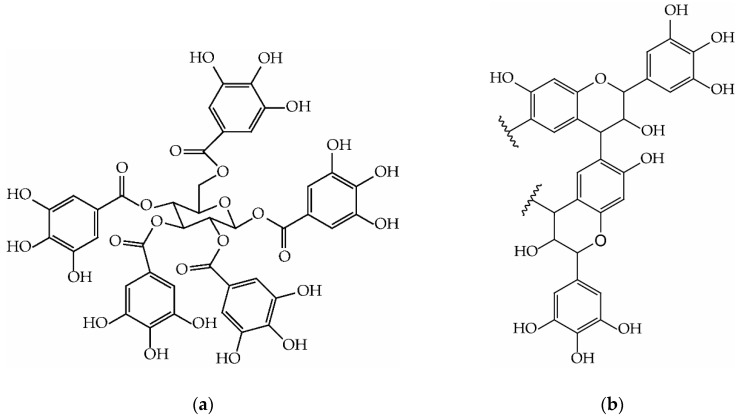
Example of polyphenols main molecules in: (**a**) Hydrolysable tannins, pentagalloylglucose (chestnut); (**b**) Condensed tannins, prorobinetinidin dimer (mimosa)

**Figure 2 biomolecules-10-00581-f002:**
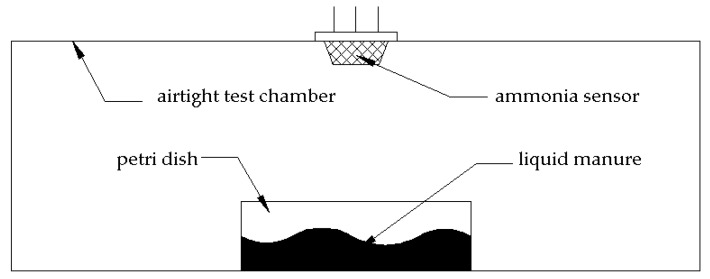
Setup of the test chamber to monitor emissions.

**Figure 3 biomolecules-10-00581-f003:**
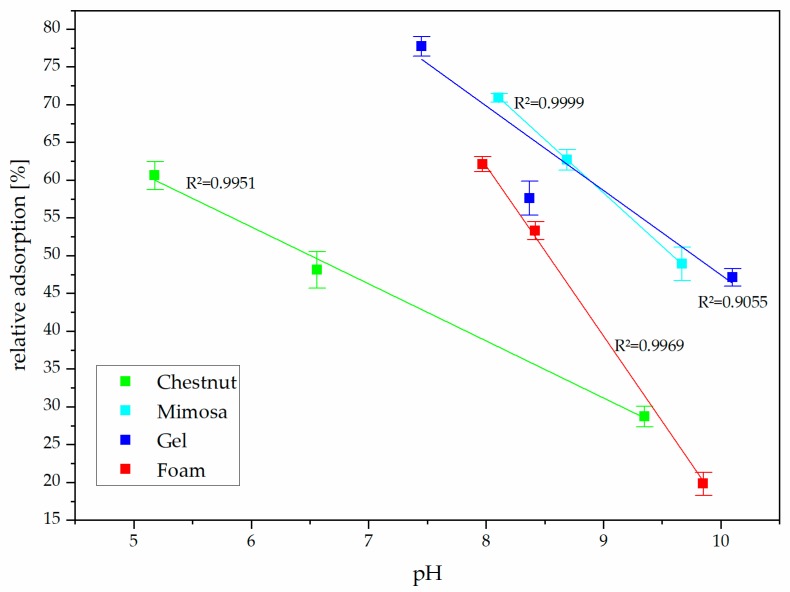
Influence of pH and tannin-based adsorbents on ammonia adsorption capacity.

**Figure 4 biomolecules-10-00581-f004:**
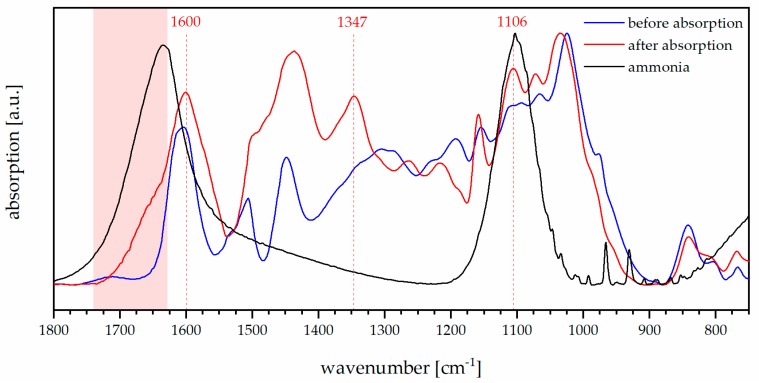
FTIR spectra of mimosa tannin before and after ammonia adsorption experiment.

**Figure 5 biomolecules-10-00581-f005:**
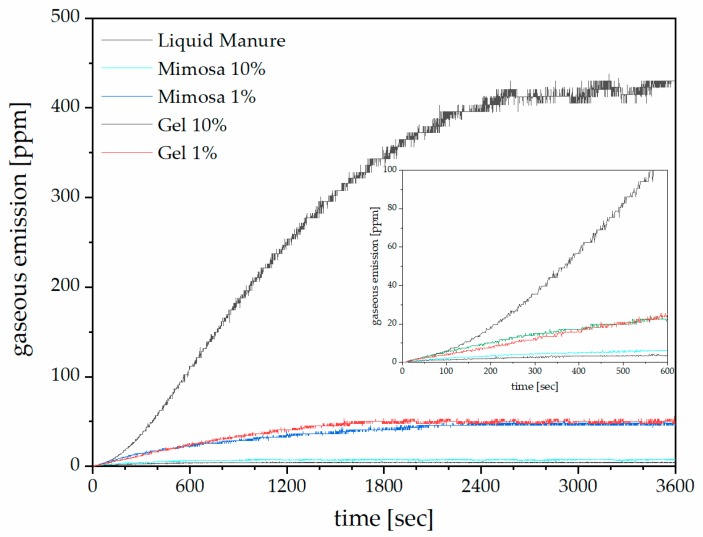
Evolution of NH_3_ in liquid manure slurry with and without tannin-based adsorbents.

**Table 1 biomolecules-10-00581-t001:** Ammonia emission from aqueous ammonia solution with and without addition of tannin-based adsorbents (standard deviation in brackets).

Adsorbent	Concentration [%]	pH	Absolute Emission [ppm]	Relative Adsorption [%]
NH_4_OH reference		10.78 (0.05)	15.80 (0.51)	00.0 (3.23)
Chestnut tannin	1	9.35 (0.23)	11.27 (0.21)	28.7 (1.34)
	5	8.56 (0.12)	8.20 (0.38)	48.1 (2.39)
	10	7.18 (0.09)	6.23 (0.29)	60.6 (1.84)
Mimosa tannin	1	9.67 (0.15)	8.07 (0.35)	48.9 (2.19)
	5	8.69 (0.24)	5.89 (0.21)	62.7 (1.35)
	10	8.11 (0.13)	4.59 (0.09)	70.9 (0.58)
Tannin gel	1	10.10 (0.11)	8.36 (0.18)	47.1 (1.15)
	5	8.37 (0.09)	6.70 (0.36)	57.6 (2.26)
	10	7.45 (0.25)	3.52 (0.20)	77.7 (1.28)
Tannin-foam powder	1	9.85 (0.37)	12.67 (0.24)	19.8 (1.54)
	5	8.42 (0.28)	7.38 (0.19)	53.3 (1.20)
	10	7.97 (0.16)	5.99 (0.16)	62.1 (1.00)

**Table 2 biomolecules-10-00581-t002:** Ammonia emission from cattle manure slurry with and without addition of tannin-based adsorbents (standard deviation in brackets).

Adsorbent	Concentration [%]	pH	Absolute Emission [ppm]	Relative Adsorption [%]
Manure reference		7.66 (0.32)	430.30 (20.1)	00.0 (4.67)
Chestnut tannin	1	7.12 (0.20)	180.30 (3.38)	58.1 (0.79)
	5	6.92 (0.13)	54.70 (7.89)	87.3 (1.83)
	10	6.61 (0.11)	44.80 (5.87)	89.6 (1.36)
Mimosa tannin	1	7.54 (0.20)	46.90 (6.05)	89.1 (1.41)
	5	7.30 (0.21)	11.50 (2.63)	97.3 (0.61)
	10	6.92 (0.15)	7.30 (1.96)	98.3 (0.46)
Tannin gel	1	7.39 (0.20)	50.10 (4.29)	88.3 (0.99)
	5	6.72 (0.21)	47.90 (4.37)	88.9 (1.01)
	10	6.27 (0.12)	4.40 (0.6)	99.0 (0.15)
Tannin-foam powder	1	7.43 (0.25)	191.00 (9.52)	55.5 (2.21)
	5	6.86 (0.11)	15.70 (3.21)	96.3 (0.75)
	10	6.17 (0.12)	10.80 (2.85)	97.5 (0.66)

**Table 3 biomolecules-10-00581-t003:** Parameters for the logistic modelling of gaseous emissions from liquid manure.

Parameter	Liquid Manure	Mimosa 10%	Mimosa 1%	Tannin Gel 10%	Tannin Gel 1%
A_1_	8.243	0.857	2.377	0.612	4.552
A_2_	468.000	7.457	56.514	4.593	52.093
x_0_	1120.041	265.630	867.117	269.066	698.098
p	2.156	1.723	1.285	1.716	2.312
R²	0.99996	0.94723	0.98583	0.91245	0.98183

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
