# Peer review of "Mitigation of Ammonia Emissions from Cattle Manure Slurry by Tannins and Tannin-Based Polymers"

_biomolecules, 2020, doi:10.3390/biom10040581_

Round 1

Reviewer 1 Report

The manuscript describes the efficacy of tannins on the reduction of ammonia and other gases from an ammonia solution and a liquid cattle manure. While the manuscript provides some novel aspects, the focus is on the mitigation of the ammonia emission by tannins, which is not new. Several studies report about the ammonia-reducing effects of tannins. The authors are encouraged to focus on structure-activity relationship to identify most promising tannins, which would increase the originality of their work.

Please explain why the 4 tannin samples were selected.

Adsorption and absorption are used interchangeably, please use appropriate term.

Line 206/207: please explain parameters of the equation.

Line 189 kinetics

Author Response

Dear Editor, Dear Reviewers,

Thank you very much for the commentaries on our paper “Mitigation of ammonia emission from cattle manure slurry by tannins and tannin-based polymers”. We have overworked the article according to the suggestions of the three reviewer and we are convinced that the present version is sensibly improved.

Reviewer 1

We thank the reviewer for the very positive feedback and suggestions for future work/ improvement of the current one.

The manuscript describes the efficacy of tannins on the reduction of ammonia and other gases from an ammonia solution and a liquid cattle manure. While the manuscript provides some novel aspects, the focus is on the mitigation of the ammonia emission by tannins, which is not new. Several studies report about the ammonia-reducing effects of tannins. The authors are encouraged to focus on structure-activity relationship to identify most promising tannins, which would increase the originality of their work.

Thank you very much for the positive feedback, we are going to take in account the study on different tannins for our future work.

Please explain why the 4 tannin samples were selected.

Thanks for the suggestion: We have added the reasons for each type of adsorbent in section 2.2

Adsorption and absorption are used interchangeably, please use appropriate term.

Yes, that’s true. The terms “absorption” were changed to “adsorption”, as the chemical surface interaction seems more appropriate for the phenomena we described.

Line 206/207: please explain parameters of the equation.

Fine. Labelling of the parameters for equation 1 was added.

Line 189 kinetics

Sorry. Spelling has been corrected

Reviewer 2 Report

Although in principle interesting, in my opinion the manuscript presents some critical points of concern so I can not recommend its publication.

  • First, why use tannins, which are high-value compounds, instead of low-cost and largely available agri-food byproducts which could absorb ammonia with equal or even higher efficiency? These could include for example tannin-rich wastes such as grape pomace, nutshells, or exhausted woods from tannin extraction.
  • Moreover, why the proposed approach should be preferred over the others listed on page 2?
  • More details on the composition of tannin gel and tannin-foam powder, concerning for example the kind and percentage amount of tannins, should have been provided to allow for a better comparison of the obtained results.
  • Lines 186-188: in my opinion this is a very important point. Maybe authors could have demonstrated specific ammonia absorption by FT-IR as in the case of mimosa tannins/ammonia solution.
  • Some sentences seem to be not very clear: lines 150-153, 161-165, and 197-202.
  • Lines 207-217: in my opinion this section is somewhat obscure, since the meaning of the parameters listed in Table 3 seems to be not clearly explained.

Some other minor points:

  • Figure 1: the main structural units specific of mimosa and chestnut tannins should have been shown.
  • Table 1: standard deviation relative to pH values should have been added.
  • Table 2: a comment should have been added to compare the data with those reported in Table 1 in terms of relative efficacy of the different tannins and tannin-based materials.

Author Response

Dear Editor, Dear Reviewers,

Thank you very much for the commentaries on our paper “Mitigation of ammonia emission from cattle manure slurry by tannins and tannin-based polymers”. We have overworked the article according to the suggestions of the three reviewer and we are convinced that the present version is sensibly improved.

Reviewer 2

Despite, this reviewer, being the more critical, we would like to take this occasion to highlight his/her intense and appropriate work. Not often the commentaries are so constructive and therefore we have done our best in order to clarify every single suggestion presented. This approach to reviewing, sometimes is given for granted, but to my experience is not more so, therefore I personally want to thank you for believing in a constructive critical scientific approach. I wish we will have you as reviewer for our future papers too.

Although in principle interesting, in my opinion the manuscript presents some critical points of concern so I can not recommend its publication.

First, why use tannins, which are high-value compounds, instead of low-cost and largely available agri-food by-products which could absorb ammonia with equal or even higher efficiency? These could include for example tannin-rich wastes such as grape pomace, nutshells, or exhausted woods from tannin extraction.

We agree with the reviewer that tannins are indeed a valuable resource and other sources can be used. Our goal was to firstly show the possibility of different classes of tannin (condensed and hydrolysable) and secondly present an alternative end-of-life use for tannin-based products (e.g. foams) which will be abundantly available in the future and only limited study on recycling them has been published so far.

This work represents the first approach to this problem, and we believe it could be a good idea for future studies to consider the use of spent wood and other by-products as ammonia scavengers.

Moreover, why the proposed approach should be preferred over the others listed on page 2?

The idea we present in this work started for finding alternative end-life solutions for the tannin foams which could be the material of the future for building insulation purposes.  We have discovered that this material has good adsorption properties against pollutant so we wanted to test its effect as mitigation agent for ammonia in cattle slurry.

The mitigation effect is taking place almost immediately after the tannin is added and in addition to other methods, especially lowering the pH by addition of sulfuric acid, tannin provides a much saver alternative for the user.

More details on the composition of tannin gel and tannin-foam powder, concerning for example the kind and percentage amount of tannins, should have been provided to allow for a better comparison of the obtained results.

Correct. Formulation and production conditions for the tannin-furanic polymer and the tannin gel were added in section 2.2

Lines 186-188: in my opinion this is a very important point. Maybe authors could have demonstrated specific ammonia absorption by FT-IR as in the case of mimosa tannins/ammonia solution.

We agree with you completely. The problem is that the FT-IR spectra of manure is very complex and it was not possible to clearly identify the peaks belonging to ammonia (neither in pure manure nor in the spectra of the tannin after the adsorption) therefore we decided to focus on the ammonia absorption, which takes place and can be detected.

Some sentences seem to be not very clear: lines 150-153, 161-165, and 197-202.

We tought that these sentences were understandable. However, we have made some style modification in order to render the concepts clearer.

Lines 207-217: in my opinion this section is somewhat obscure, since the meaning of the parameters listed in Table 3 seems to be not clearly explained.

Thanks for the remark: Labelling for the parameter used in equation 1 has been added to clarify its meaning

Some other minor points:

Figure 1: the main structural units specific of mimosa and chestnut tannins should have been shown.

We agree. Then we changed the more general structure with more punctual ones selecting  pentagalloylglucose for the chestnut and prorobinetinidin unit for the mimosa tannins.

Table 1: standard deviation relative to pH values should have been added.

Sorry, we forgot to report it. We have added the standard deviation to pH values for both, table 1 and table 2

Table 2: a comment should have been added to compare the data with those reported in Table 1 in terms of relative efficacy of the different tannins and tannin-based materials.

Correct. A paragraph focusing on the relative change between different concentration of adsorbents and differences between ammonia and manure has been added after the data.

Reviewer 3 Report

This is a very interesting manuscript that has practical applications for livestock producers.

My comments are primarily editorial in nature 

Line 38 Wang needs a referrence

41 In fact,....

42 ...also the ozone...

45-46 please reword

50 feed not food

52 lower crude protein

54 stockpile of what?

72 tannins are getting more attractive to do what?

75 and as replace? 

85 clarified

141 reword

Author Response

Dear Editor, Dear Reviewers,

Thank you very much for the commentaries on our paper “Mitigation of ammonia emission from cattle manure slurry by tannins and tannin-based polymers”. We have overworked the article according to the suggestions of the three reviewer and we are convinced that the present version is sensibly improved.

Reviewer 3

We thank the reviewer for this positive feedback and the editorial corrections suggested.

This is a very interesting manuscript that has practical applications for livestock producers.

Thanks for the kind feedback.

My comments are primarily editorial in nature

Line 38 Wang needs a reference

Indeed. We added the related reference.

41 In fact, ...

42 ...also the ozone...

45-46 please reword

50 feed not food

52 lower crude protein

54 stockpile of what?

72 tannins are getting more attractive to do what?

75 and as replace?

85 clarified

141 reword

Thanks for the suggestions, We clarified and corrected the text in every sections.

Round 2

Reviewer 1 Report

The major concern of the reviewer about the originality of the work was recognized and confirmed by the authors but did not lead to any improvements of the manuscript. Even without any additional experiments the novelty could have been enhanced as suggested by the reviewer.

Anyways, all minor remarks were considered by the authors.

The reviewer is not fully convinced about the scientific novelty of the manuscript but leaves it up to the other reviewers and the editor whether the manuscript can be recommended for publication in its present form. For sure, the manuscript provides some practical relevance, as also reported elsewhere.

Reviewer 2 Report

Though I was not in favor of manuscript publication, authors have managed to convince me about the aim of the paper and the perspectives opened by their research. All my other concerns were also sufficiently addressed so I can now recommend publication.